# Impact of Bone-Modifying Agents on Post-Bone Metastasis Survival Across Cancer Types

**DOI:** 10.3390/curroncol32010042

**Published:** 2025-01-15

**Authors:** Hironari Tamiya, Kazumi Nishino, Yuji Kato, Reina Nakahashi-Kato, Yurika Kosuga-Tsujimoto, Shota Kinoshita, Rie Suzuki, Makiyo Watanabe, Toru Wakamatsu, Shigeki Kakunaga, Satoshi Takenaka

**Affiliations:** 1Department of Rehabilitation, Osaka International Cancer Institute, Osaka 541-8567, Japan; yuuji.katou@oici.jp (Y.K.); nakahashir@opho.jp (R.N.-K.); kosugay@opho.jp (Y.K.-T.); kinoshitash@opho.jp (S.K.); 2Department of Orthopedic Surgery, Osaka International Cancer Institute, Osaka 541-8567, Japan; rie.suzuki@oici.jp (R.S.); makiyo.watanabe@oici.jp (M.W.); tooru.wakamatsu@oici.jp (T.W.); shigeki.kakunaga@oici.jp (S.K.);; 3Department of Thoracic Oncology, Osaka International Cancer Institute, Osaka 541-8567, Japan; kazumi.nishino@oici.jp

**Keywords:** bone-modifying agent, bone metastasis, survival, cancer, treatment

## Abstract

Background: Bone metastasis is associated with a poor prognosis. Bone-modifying agents (BMA) are commonly used for the prevention or treatment of skeletal-related events (SRE) in patients with bone metastasis; however, whether or not treatment with BMA improves survival remains unclear. In this study, we investigated whether BMA was involved in post-bone metastasis survival. Methods: A total of 539 cancer patients were retrospectively analyzed to identify significant independent factors in post-bone metastasis survival. Results: Among the overall population, patients with the following cancers had a median survival longer than 24 months: thyroid, 97.2 months; breast, 51.5 months; prostate, 47.2 months; and kidney, 38.8 months. In contrast, median post-bone metastasis survival was significantly shorter in gastrointestinal (GI) (6.5 months), head and neck (6.3 months), and urinary tract (3.4 months) cancers. In non-small cell lung cancer (NSCLC), the log-rank test demonstrated that the epidermal growth factor receptor (EGFR) mutation was a significant factor for post-bone metastasis survival: EGFR mutation (−) *n* = 67, median post-bone metastasis survival 11.5 months (95% CI: 6.0–15.2); EGFR mutation (+) *n* = 39, median post-bone metastasis survival 28.8 months (95% CI: 18.1–35.7) (*p* < 0.05). Intriguingly, treatment with BMA was a significant positive prognostic factor: BMA (−) *n* = 203, median post-bone metastasis survival 7.8 months (95% CI: 5.8–12.5); BMA (+) *n* = 336, median post-bone metastasis survival 21.9 months (95% CI: 16.1–26.4) (*p* < 0.001). Moreover, the Cox proportional hazards model showed that this was particularly evident in cancer types with poor prognosis such as GI cancer (hazard ratio [HR]: 0.62, 95% CI: 0.40–0.95; *p* < 0.05) and NSCLC without the epidermal growth factor receptor (EGFR) mutation (HR: 0.56, 95% CI: 0.34–0.91; *p* < 0.05). Conclusions: Treatment with BMA is recommended not only for the prevention and/or treatment of SRE, but also may have a positive impact on post-bone metastasis survival, particularly in cancers with typically poor post-bone metastasis survival such as GI cancer and NSCLC without the EGFR mutation.

## 1. Introduction

Bone metastases can develop in patients with all types of cancer at advanced stages. As previously reported, the incidence of bone metastasis varies according to the cancer type; lung, prostate, breast, and kidney cancers have high incidences [1,2]. Reports have also shown that bone metastasis is associated with poor prognosis [3,4,5,6]. For example, in prostate cancer, 5-year survival rates were 56% and 3% in patients without and with bone metastases, respectively [3]. Moreover, in men with bone metastasis minus skeletal related events (SRE), hazard ratios (HR) for mortality were 6.6 (95% CI: 6.4–6.9), and in men with bone metastasis plus SRE they were 10.2 (95% CI: 9.8–10.7) [4]. Similarly, in breast cancer, the 5-year survival rates were 75.8% and 8.3% in patients without and with bone metastases, respectively [5]. Bone metastasis impairs physical function and quality of life (QOL) in patients with cancer due to multiple complications such as pathological fractures, paralysis due to spinal cord compression, and disuse syndrome. Hence, proper treatment of bone metastasis is essential to achieve a good QOL in patients with cancer [7].

Impairment of physical function caused by bone metastasis leads to decline of performance status (PS) that is profoundly related to prognosis. A previous study mentioned that bone metastasis is a significant independent negative predictive factor for overall survival (OS) in lung cancer patients with mutated as well as wild-type epidermal growth factor receptor (EGFR) whereas lung or brain metastasis was not involved in OS, which is also suggesting the importance of managing bone metastasis [6]. Similarly, bone metastasis negatively affects prognosis in other types of cancer including thyroid, colorectal, or kidney cancer [8,9,10].

From another aspect, independence of activity of daily living (ADL) is essential for good QOL. Loss of independence is a worldwide social problem. Disability in ADL has been studied in the field of aging [11]. In addition, in the field of oncology, cancer patients also have difficulty in ADL due to physical dysfunction [12]. A previous study shows that cancer promotes locomotive syndrome, leading to poorer QOL [13]. We thus are facing social issues about not only aging but also cancer-related weakness.

When it comes to treatment for lung cancer, antitumor agents have been developed for lung cancer in the past decades. Molecular-targeted drugs, including an EGFR tyrosine kinase inhibitor, drastically improved OS in patients with lung cancer in whom a specific driver mutation is found [14]. Moreover, immune checkpoint inhibitors (ICI) also prolong OS as compared with conventional chemotherapy [15].

Bone-modifying agents (BMA) have been commonly used in patients with bone metastasis to prevent or treat SRE. However, BMA treatment cannot necessarily be possible because of side effects such as hypocalcemia, atypical femoral fractures (AFFs), and medication-related osteonecrosis of the jaw (MRONJ) [16,17]. To prevent hypocalcemia, administration of vitamin D and calcium is required. Once AFFs occur, nonunion rate is high because of BMA-induced inactive bone formation [18]. MRONJ is also difficult to treat and sometimes hinders the use of chemotherapy [19]. Another point to consider is that the effect of BMA varies according to types of cancer [20]. Specifically, breast, prostate, and multiple myeloma can be treated by BMA every 12 weeks, although BMA should be administered every 4 weeks in the other types of cancer, which indicates the effect of BMA may be different between various types of cancer. Moreover, it is unclear whether the management of bone metastasis improves survival [21,22,23,24]. In the present study, we investigated whether BMA improved post-bone metastasis survival and analyzed the difference between various types of cancer.

## 2. Materials and Methods

This study included 539 patients with cancer and bone metastasis who underwent rehabilitation at the Osaka International Cancer Institute between 2018 and 2020. The date of bone metastasis was obtained from medical records.

The cancer types were classified as non-small cell lung cancer (NSCLC), breast, prostate, kidney, colorectal, bone and soft tissue, urinary tract, pancreas, esophagus, head and neck, liver, small cell lung cancer (SCLC), stomach, skin, thyroid, uterus, and biliary tract. The urinary tract included the bladder, renal pelvis, ureters, and urethra. Uterine cancer comprised the uterine cervix, uterine corpus, and uterine sarcoma. The biliary tract included the gallbladder, bile duct, and the duodenal papillae. Others include thirteen cancers of unknown primary origin, one anal canal cancer, one mesothelioma, one hemangiopericytoma, one malignant paraganglioma, one neuroendocrine tumor, one olfactory neuroblastoma, one anal fistula cancer, and one penile cancer.

BMA treatments were defined as the administration of zoledronic acid or denosumab at least twice every 4 weeks after bone metastasis. The use and selection of BMA (zoledronic acid or denosumab) depend on physicians’ preferences. BMA is basically initiated immediately after detection of the first bone metastasis and subsequent dentist consultation.

Rehabilitation is performed according to general condition as well as condition of bone metastasis. Rehabilitation for impaired extremity or trunk such as painful bone metastasis or impending fracture is prohibited whereas rehabilitation for painless and stable bone metastasis can be conducted. Physiatrists and orthopedists judge whether rehabilitation should be done or not before initiation of rehabilitation. Rehabilitation as much as possible could inhibit disuse syndrome to improve or maintain PS, potentially leading to better prognosis.

The Kaplan–Meier method was used to calculate post-bone metastasis survival. The log-rank test was used for univariate analysis. Multivariate analysis was conducted using the Cox proportional hazards model. Associations of post-bone metastasis survival with age, BMA, cancer type, driver mutation, surgery, and/or radiotherapy for bone metastasis were retrospectively analyzed. Differences were considered statistically significant at *p* < 0.05. Statistical analyses were performed using the EZR (64 bit) [25].

To choose factors for multivariate analysis, univariate analysis was first conducted. Significant factors indicated in univariate analysis were thereafter investigated using multivariate analysis.

This study was conducted following the Declaration of Helsinki and was approved by the institutional review board of Osaka International Cancer Institute (IRB no. 22115–2 approved on 3 October 2022).

## 3. Results

Of the 539 patients, 301 were age ≥65 years (55.8%) and 238 were <65 years (44.2%). A total of 304 men (56.4%) and 235 women (43.6%) were included in this study. The primary origin types of cancer were: NSCLC, 106 (19.7%); breast, 84 (15.6%); prostate, 48 (8.9%); kidney, 34 (6.3%); colorectum, 30 (5.6%); bone and soft tissue, 27 (5.0%); urinary tract, 26 (4.8%); pancreas, 23 (4.3%); esophagus, 22 (4.1%); head and neck, 20 (3.7%); liver, 19 (3.5%); SCLC, 18 (3.3%); stomach, 18 (3.3%); skin, 14 (2.6%); thyroid, 11 (2.0%); uterus, 11 (2.0%); biliary tract, 7 (1.3%); and others, 21 (3.9%). Median post-bone metastasis survival largely varied according to cancer type: thyroid, 97.2 months (95% CI: 11.6–N/A [not applicable]); breast, 51.5 months (95% CI: 37.7–69.1); prostate, 47.2 months (95% CI: 30.2–79.3); and kidney, 38.8 months (95% CI: 18.9–85.9); all of which resulted in a longer than 24 month survival (Table 1). In the present study, median post-bone metastasis survival was 15.2 months (95% CI: 12.8–19.2) for the overall population (Figure 1A). We also analyzed the relationships between post-bone metastasis survival and age, sex, and BMA levels. Age was not significantly associated with post-bone metastasis survival in the overall population: age <65, *n* = 238, median post-bone metastasis survival 15.1 months (95% CI: 12.5–19.8); age ≥65, *n* = 301, median post-bone metastasis survival 16.8 months (95% CI: 11.0–22.1) (*p* = 0.61). Sex and BMA were significant factors related to post-bone metastasis survival in the overall population in the univariate analyses: female, *n* = 235 median post-bone metastasis survival 25.7 months (95% CI: 18.4–31.0); male, *n* = 304 median post-bone metastasis survival 11.9 months (95% CI: 10.2–14.2) (*p* < 0.005); BMA (−) *n* = 203, median post-bone metastasis survival 7.8 months (95% CI: 5.8–12.5); BMA (+) *n* = 336, median post-bone metastasis survival 21.9 months (95% CI: 16.1–26.4) (*p* < 0.001) (Figure 1B,C). As treatment for bone metastasis, radiation was done in 382 patients; surgery was performed in 54. A total of 103 cases had neither radiation nor surgery for bone metastasis. Radiation was significantly related to poor post-bone metastasis survival: radiation, *n* = 382, median post-bone metastasis survival 12.6 months (95% CI: 10.8–15.4); surgery, *n* = 54, median post-bone metastasis survival 24.6 months (95% CI: 13.6–80.4); neither radiation nor surgery, *n* = 103, median post-bone metastasis survival 24.1 months (95% CI: 17.6–35.3) (*p* < 0.01). In addition, at the first visit to the department of rehabilitation, 457 out of 539 (84.8%) cases already had bone metastasis, whereas 82 (15.2%) patients did not. The presence of bone metastasis at the first visit of the department of rehabilitation was not associated with post-bone metastasis survival: bone metastasis at the first visit (+), *n* = 457, median post-bone metastasis survival 16.2 months (95% CI: 13.3–20.1); bone metastasis at the first visit (−), *n* = 82, median post-bone metastasis survival 10.4 months (95% CI: 5.4–26.0) (*p* = 0.09). Regarding metastasis other than bone, at the first visit to the department of rehabilitation, 204 (37.8%) patients had no metastasis other than bone; 75 (13.9%) had lung metastasis; 70 (13.0%) had liver metastasis; 67 (12.4%) had brain metastasis; 62 (11.5%) had dissemination; and 61 (11.3%) patients had some metastases in more than two sites. Importantly, metastasis other than bone was significantly related to post-bone metastasis survival: metastasis other than bone (−), *n* = 204, median post-bone metastasis survival 23.3 months (95% CI: 11.8–30.2); metastasis other than bone (+), *n* = 335, median post-bone metastasis survival 14.1 months (95% CI: 11.9–17.6) (*p* < 0.005) (Figure 1D). Especially, as compared to metastasis other than bone (−), metastasis in more than two sites and dissemination were significantly associated with poor post-bone metastasis survival (brain metastasis, *n* = 67, hazard ratio [HR] 1.06, 95% CI: 0.74–1.50, *p* = 0.76; lung metastasis, *n* = 75, HR 1.26, 95% CI: 0.89–1.78, *p* = 0.18; liver metastasis, *n* = 70, HR 1.37, 95% CI: 0.99–1.90, *p* = 0.053; metastasis in more than two sites, *n* = 61, HR 1.60, 95% CI: 1.15–2.24, *p* < 0.01; dissemination, *n* = 62, HR 1.92, 95% CI: 1.38–2.67, *p* < 0.001) (Table 2). Of note, multivariate analysis eliminated the significance of the presence of metastasis other than bone, treatment for bone metastasis and sex, but not BMA (Table 1).

We further aimed to clarify the effects of BMA on post-bone metastasis survival. First, gastrointestinal (GI) cancers were examined and results showed that median post-bone metastasis survival was 6.5 months (95% CI: 4.0–8.9), *n* = 119 (Figure 2A). Notably, treatment with BMA was significantly correlated with better post-bone metastasis survival in GI cancers: BMA (−) *n* = 56, median post-bone metastasis survival 3.2 months (95% CI: 1.8–5.9); BMA (+) *n* = 63, median post-bone metastasis survival 8.9 months (95% CI: 6.1–12.6) (*p* < 0.05) (Figure 2B). Cox proportional hazard model indicated that cancer type was not significantly involved in post-bone metastasis survival (Table 3). By log-rank tests, age, sex, and metastasis other than bone were also unrelated to post-bone metastasis survival: age < 65, *n* = 55, median post-bone metastasis survival 7.6 months (95% CI: 3.4–10.9); age ≥ 65, *n* = 64, median post-bone metastasis survival 5.9 months (95% CI: 3.3–11.0) (*p* = 0.49); female, *n* = 37, median post-bone metastasis survival 6.9 months (95% CI: 3.2–10.9); male, *n* = 82, median post-bone metastasis survival 6.1 months (95% CI: 3.7–10.8) (*p* = 0.35); metastasis other than bone (−), *n* = 39, median post-bone metastasis survival 3.0 months (95% CI: 2.0–6.9); metastasis other than bone (+), *n* = 80, median post-bone metastasis survival 7.9 months (95% CI: 5.5–12.2) (*p* = 0.14). From several univariate analyses, BMA was the only significant factor for post-bone metastasis survival in the present study.

Next, we focused on NSCLC (*n* = 106), which included eighty-one (76.4%) adenocarcinomas, twenty-three (21.7%) squamous cell carcinomas, and two (1.9%) large cell carcinomas. The epidermal growth factor receptor (EGFR) mutation was observed in 39 (36.8%) patients. Median post-bone metastasis survival in NSCLC was 17.6 months (95% CI: 11.6–24.9) (Figure 3A). The EGFR mutation was a positive prognostic marker of post-bone metastasis survival: EGFR mutation (−) *n* = 67, median post-bone metastasis survival 11.5 months (95% CI: 6.0–15.2); EGFR mutation (+) *n* = 39, median post-bone metastasis survival 28.8 months (95% CI: 18.1–35.7) (*p* < 0.05) (Figure 3B). Furthermore, treatment with BMA was also a positive prognostic indicator: BMA (−) *n* = 39, median post-bone metastasis survival 10.7 months (95% CI: 4.9–22.0); BMA (+) *n* = 67, median post-bone metastasis survival 24.9 months (95% CI: 13.5–31.0) (*p* < 0.05) (Figure 3C). In subgroup analyses, the EGFR mutation (−) group confirmed the significance of BMA for better post-bone metastasis survival: BMA (−) *n* = 28, median post-bone metastasis survival 7.2 months (95% CI: 3.4–12.9); BMA (+) *n* = 39, median post-bone metastasis survival 13.6 months (95% CI: 6.0–27.1) (*p* < 0.05) (Figure 3D). However, the EGFR mutation (+) group did not: BMA (−) *n* = 11, median post-bone metastasis survival 24.1 months (95% CI: 7.0–N/A); BMA (+) *n* = 28, median post-bone metastasis survival 31.0 months (95% CI: 17.6–36.5) (*p* = 0.49) (Figure 3E). The multivariate analysis consistently showed that both EGFR mutations and BMA levels were independent prognostic factors (Table 4).

Finally, we attempted to identify the types of cancers in which BMA significantly improved post-bone metastasis survival. Results showed that BMA did not significantly improve post-bone metastasis survival in cancers with a better prognosis than 24 months including breast, kidney, prostate, and thyroid cancers: BMA (−) *n* = 39, median post-bone metastasis survival 47.2 months (95% CI: 25.7–97.2); BMA (+) *n* = 138, median post-bone metastasis survival 42.0 months (95% CI: 33.9–67.3) (*p* = 0.69). But, significantly longer post-bone metastasis survival was observed with BMA treatment in the counterpart (cancers with less than 24 months prognosis for survival): BMA (−) *n* = 164, median post-bone metastasis survival 5.5 months (95% CI: 3.5–7.4); BMA (+) *n* = 198, median post-bone metastasis survival 11.7 months (95% CI: 8.3–13.5) (*p* < 0.005) (Figure 4A,B).

## 4. Discussion

This study demonstrated that treatment with BMA was a positive prognostic factor for post-bone metastasis survival, especially in cancers with a poor prognosis after bone metastasis (which did not include breast, kidney, prostate, and thyroid cancers).

An earlier report suggested a positive effect in patients treated with the BMAs docetaxel and zoledronic acid. These treatments provided better bone-metastasis progression-free survival (docetaxel with zoledronic acid 9.0 vs. docetaxel alone; 6.0 months; *p* < 0.05) and overall survival (OS) (docetaxel with zoledronic acid, 19.0 vs. docetaxel alone; 15.0 months; *p* < 0.05) than that provided by zoledronic acid alone in castration-resistant prostate cancer [26], although another paper showed no significant effect [27]. As recommended in clinical practice, BMA can reduce SRE, which may help maintain or improve performance status. Therefore, BMA is recommended for cancer patients with bone metastasis, although careful consideration should be given to AFF or MRONJ. Reports show that the incidence of AFF or MRONJ is less than 1% with BMA treatment [28,29].

Advancements in cancer treatment, including surgery, chemotherapy, and radiotherapy, have improved the survival of patients with several types of cancer [30]. Similarly, post-bone metastasis survival might also improve due to advances in cancer treatment, although previous reports have not yet shown a positive effect, and patients with bone metastasis have demonstrated poor survival [31,32]. Molecularly-targeted drugs have been associated with prolonged survival in patients with lung cancer and post-bone metastasis [33]. In line with these findings, the results of our study demonstrate that the presence of EGFR mutations is significantly related to better post-bone metastasis survival. Physicians should be mindful that bone metastasis and primary lung lesions can be controlled by molecularly targeted drugs, leading to improved post-bone metastasis survival.

Some plausible reasons for why we observed longer post-bone metastasis survival with BMA treatment can be proposed. The characteristics of individual cancers, including aggressiveness of the tumor, sensitivity to antitumor agents, or radiotherapy, may play a role. For example, in most cases, thyroid cancer is slow-growing [34], suggesting that the expected prognosis of thyroid cancer might be better than that of more aggressive tumors, even after bone metastasis occurs. Additionally, bone formation is occasionally observed with molecularly-targeted drugs in NSCLC with EGFR mutations [35], indicating that BMA may not be required when bone formation is sufficient without BMA treatment. Most cancers with poor post-bone metastasis survival appear to be less sensitive to antitumor agents or radiotherapy. In contrast, breast and prostate cancers tend to be sensitive to hormone therapy, indicating that BMA treatment may be less effective for post-bone metastasis survival in these cancers, even though BMA is essential for preventing and/or treating SRE [7].

Independence in activities of daily living (ADL) is crucial for a good QOL and the loss of independence is a global social problem. Physical disabilities have been a focus of research on aging [11]. In the oncology field, cancer patients have also experienced difficulties with ADL due to physical dysfunction [12]. A previous study showed that cancer promotes locomotive syndrome, leading to poor QOL [13]. Consequently, we are facing social issues regarding not only aging but also cancer-related weaknesses. Additionally, the impairment of physical function caused by bone metastasis leads to a decline in performance status, which is strongly related to prognosis. For instance, a previous study reported that bone metastases were a significant independent negative predictive factor for OS in lung cancer patients with mutated and wild-type EGFR, whereas lung or brain metastases were not associated with OS, highlighting the importance of managing bone metastasis [6].

Treatment guideline suggests the use of BMA for bone metastasis, although there are no randomized data to guide whether all patients with bone metastases should initiate a BMA as soon as bone metastases are diagnosed [17]. As our data show BMA may improve post-bone metastasis survival, early initiation of BMA immediately after detection of bone metastasis seems preferred.

In regards of survival improvement by BMA, there are some plausible underlying mechanisms. One of the most plausible reasons is the involvement of bone metastases in poor prognosis, as mentioned above. In several types of cancers, the presence of bone metastasis is significantly related to poor prognosis. Bone metastasis causes SRE, leading to a decline in PS which is strongly associated with prognosis. BMA is well known to reduce SRE, potentially resulting in PS improvement/maintenance and better prognosis. Moreover, treatment guidelines indicate algorism for use of BMA according to the effect of chemotherapy, suggesting discontinuation after 24 months when good response is achieved [17], suggesting more importance of BMA in chemotherapy-resistant cancers with bone metastasis. This may be able to explain a positive impact of BMA on GI cancer and NSCLC without EGFR mutation in which prognosis is not so good due to insufficient effectiveness of chemotherapy.

Regarding diagnosis of bone metastasis, whole-body diffusion-weighted MRI (WB-DWI) with background body suppression (DWIBS) is reported effective for detecting bone metastasis [36]. In addition, a previous report mentioned that metastatic tumor volume based on apparent diffusion coefficient values evaluated by WB-DWI is significantly associated with cancer-specific survival and might be a potential prognostic imaging biomarker for castration-resistant prostate cancer [37].

The present study had some limitations. First, the study may have had a selection bias since all included patients underwent rehabilitation at our institute, indicating that they were less healthy than those who were outpatients and visited the hospital on their own. Furthermore, the number of events was not large in the multivariate analysis; thus, the statistical power may not have been sufficient to reach a solid conclusion. Further investigation is warranted to address these limitations.

One of the major strengths of this study is that BMA may have a potential to improve post-bone metastasis survival. BMA is commonly used for prevention or treatment of SRE, not for improvement of post-bone metastasis survival. The novelty of this study is to highlight the role of BMA on post-bone metastasis survival. In this regard, this study may shed light on the possibility of BMA for better prognosis. Another strength is to point out the fact that bone metastasis should not be underestimated in cancers with poor prognosis such as GI cancers where bone metastasis is not so common as those in lung, breast, and prostate cancers.

The message from this study is that BMA may be effective not only for prevention of SRE but also for improvement of prognosis. BMA is recommended even in cancers with poor prognosis.

## 5. Conclusions

Treatment with BMA is recommended not only for the prevention and/or treatment of SRE, but also to improve survival, particularly in cancers with typically poor post-bone metastasis survival such as GI cancer and NSCLC without the EGFR mutation. BMA should be initiated immediately after detection of the first bone metastasis considering hypocalcemia, AFF, or MRONJ as adverse effects.

## Figures and Tables

**Figure 1 curroncol-32-00042-f001:**
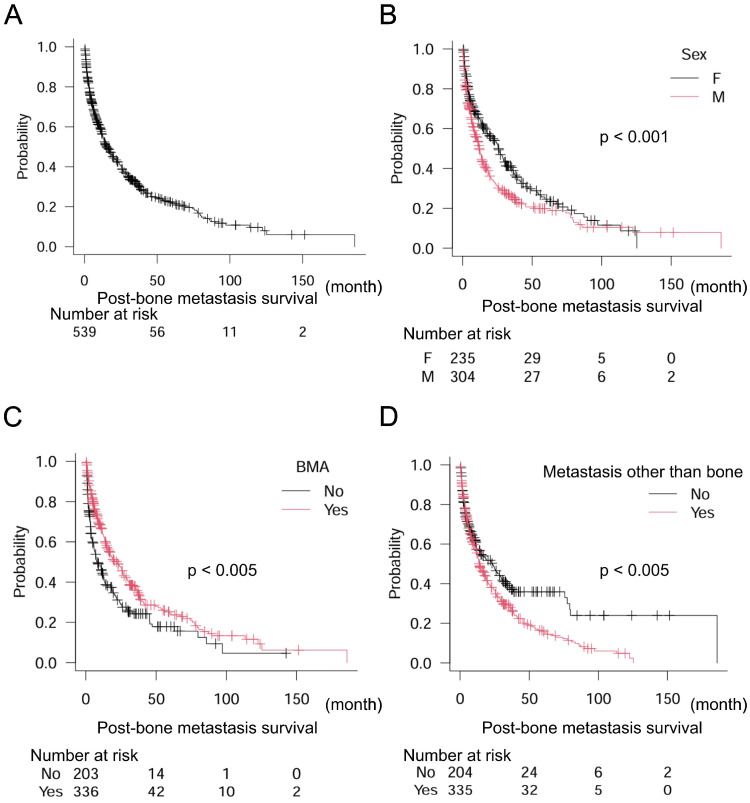
Post-bone metastasis survival in (**A**) the overall population and (**B**) dichotomized groups by sex, (**C**) dichotomized groups by BMA, and (**D**) dichotomized groups by metastasis other than bone in all 539 patients. Log-rank test is used for statistical analysis. A *p*-value of 0.05 or lower is considered statistically significant.

**Figure 2 curroncol-32-00042-f002:**
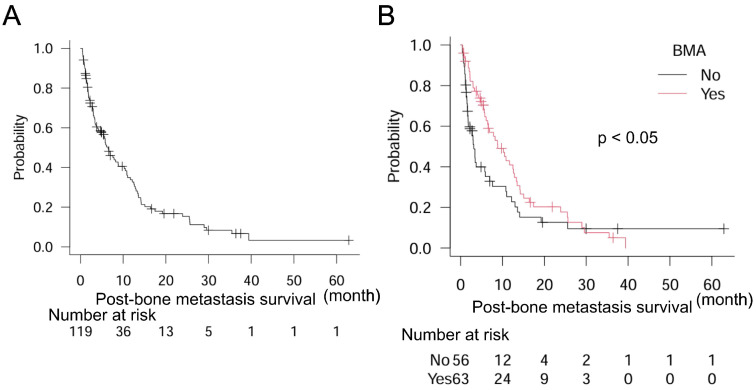
Post-bone metastasis survival in the overall population and patients with gastrointestinal cancers. (**A**) Overall population and (**B**) dichotomized groups by treatment with BMA in 119 patients with gastrointestinal cancers. Log-rank test is used for statistical analysis. A *p*-value of 0.05 or lower is considered statistically significant.

**Figure 3 curroncol-32-00042-f003:**
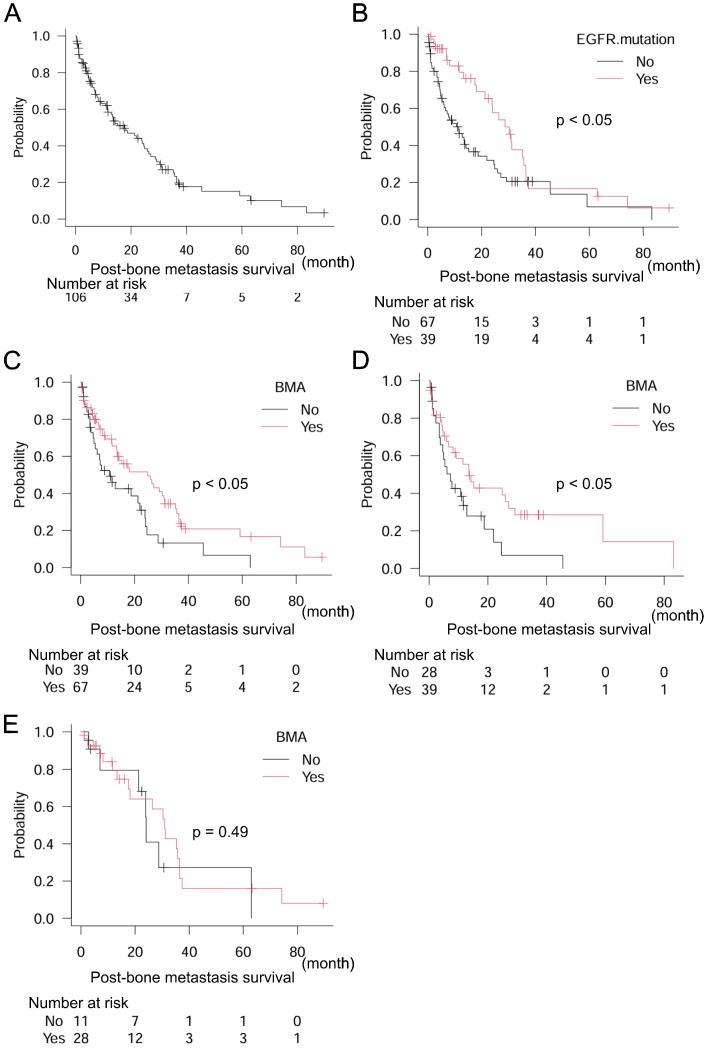
Post-bone metastasis survival in the overall population and patients with NSCLC and EGFR mutation. (**A**) Overall population, (**B**) dichotomized groups by EGFR mutation, and (**C**) dichotomized groups by treatment with BMA in 106 patients with NSCLC, (**D**) dichotomized groups by treatment with BMA in patients without the EGFR mutation (−), and (**E**) dichotomized groups by treatment with BMA in patients with the EGFR mutation (+). Log-rank test is used for statistical analysis. A *p*-value of 0.05 or lower is considered statistically significant.

**Figure 4 curroncol-32-00042-f004:**
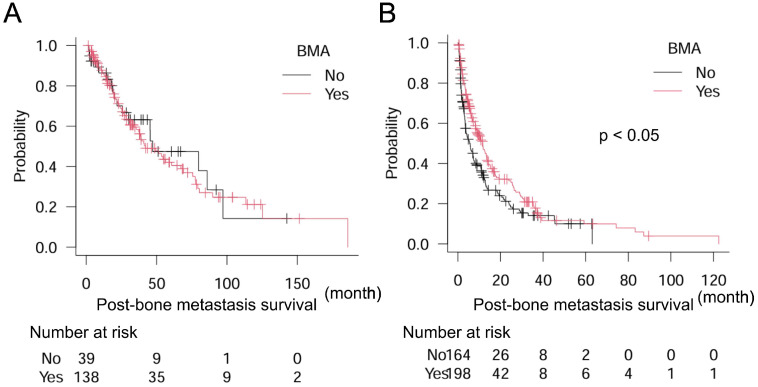
Association of BMA with post-bone metastasis survival in patients with thyroid, breast, prostate, kidney, and other cancers in (**A**) 177 patients with thyroid, breast, prostate, and kidney cancers (post-bone metastasis survival better than 24 months) and (**B**) 362 patients with the other cancers (post-bone metastasis survival poorer than 24 months). Log-rank test is used for statistical analysis. A *p*-value of 0.05 or lower is considered statistically significant.

**Table 1 curroncol-32-00042-t001:** Univariate and multivariate analyses showing median post-bone metastasis survival and associated factors according to cancer types in the overall population.

			Univariate				Multivariate		
Factor	Number	Median Post-Bone Metastasis Survival (Months)	Hazard Ratio	95% CI	*p*-Value	Hazard Ratio	95% CI	*p*-Value	
Cancer Type	Thyroid	11	97.2	0.22	0.07–0.72	<0.05	0.22	0.07–0.70	<0.05
	Breast	84	51.5	0.39	0.27–0.57	<0.001	0.45	0.30–0.66	<0.001
	Prostate	48	47.2	0.34	0.21–0.54	<0.001	0.36	0.22–0.59	<0.001
	Kidney	34	38.8	0.48	0.29–0.79	<0.005	0.47	0.28–0.78	<0.005
	NSCLC	106	17.6	Reference			Reference		
	Bone and soft tissue	27	11.9	1.04	0.61–1.76	0.89	0.93	0.54–1.59	0.78
	Liver	19	10.8	1.48	0.83–2.62	0.18	1.42	0.80–2.53	0.23
	Colorectum	30	8.8	1.56	0.96–2.54	0.07	1.44	0.88–2.35	0.15
	SCLC	18	8.6	1.44	0.78–2.65	0.25	1.23	0.66–2.28	0.52
	Stomach	18	6.9	2.26	1.29–3.95	<0.005	2.28	1.29–4.03	<0.005
	Head and neck	20	6.3	2.30	1.33–3.97	<0.005	2.23	1.29–3.85	<0.005
	Pancreas	23	5.5	2.26	1.36–3.75	<0.005	2.30	1.38–3.84	<0.005
	Skin	14	5.1	2.08	1.07–4.03	<0.05	1.67	0.85–3.28	0.13
	Biliary tract	7	4.2	2.73	1.09–6.79	<0.05	2.29	0.91–5.75	0.07
	Uterus	11	3.6	1.20	0.60–2.41	0.61	1.41	0.69–2.88	0.34
	Urinary tract	26	3.4	2.89	1.77–4.72	<0.001	2.76	1.67–4.56	<0.001
	Esophagus	22	2.9	2.94	1.76–4.90	<0.001	2.55	1.51–4.30	<0.001
	Others	21	8.3	1.11	0.65–1.89	0.70	1.06	0.62–1.81	0.83
BMA	No	203	7.8	Reference			Reference		
	Yes	336	21.9	0.64	0.51–0.79	<0.001	0.72	0.57–0.91	<0.01
Metastasis other than bone	No	204	23.3	Reference			Reference		
	Yes	335	14.1	1.40	1.12–1.75	<0.005	1.19	0.94–1.51	0.15
Sex	Female	235	25.7	Reference			Reference		
	Male	304	11.9	1.39	1.13–1.72	<0.005	1.15	0.89–1.48	0.29
Treatment for bone metastasis	No	103	24.1	Reference			Reference		
	Yes	436	13.6	1.39	1.05–1.84	<0.05	1.33	0.99–1.78	0.059

Cox proportional hazard model is used for statistical analysis. A *p*-value of 0.05 or lower is considered statistically significant. CI: confidence interval.

**Table 2 curroncol-32-00042-t002:** Univariate analysis showing median post-bone metastasis survival and association with metastasis other than bone in the overall population.

Metastatic Site	Number	Median Post-Bone Metastasis Survival (Months)	Hazard Ratio	95% CI	*p*-Value
None	204	23.3	Reference		
Brain	67	21.3	1.06	0.74–1.50	0.76
Lung	75	17.4	1.26	0.89–1.78	0.18
Liver	70	13.5	1.37	0.99–1.90	0.053
More than two sites	61	12.6	1.60	1.15–2.24	<0.01
Dissemination	62	12.2	1.92	1.38–2.67	<0.001

Cox proportional hazard model is used for statistical analysis. A *p*-value of 0.05 or lower is considered statistically significant. CI: confidence interval.

**Table 3 curroncol-32-00042-t003:** Univariate analysis of cancer type for post-bone metastasis survival in GI cancers.

Factor		Hazard Ratio	95% CI	*p*-Value
Cancer Type	Colorectum	Reference		
	Liver	0.97	0.49–1.91	0.92
	Pancreas	1.43	0.76–2.68	0.26
	Stomach	1.46	0.75–2.86	0.27
	Biliary tract	1.68	0.63–4.52	0.3
	Esophagus	1.81	0.96–3.40	0.07

Cox proportional hazard model is used for statistical analysis. A *p*-value of 0.05 or lower is considered statistically significant. CI: confidence interval.

**Table 4 curroncol-32-00042-t004:** Multivariate analysis for independent factors (BMA and EGFR mutation) related to post-bone metastasis survival in NSCLC. Cox proportional hazard model is used for statistical analysis. A *p*-value of 0.05 or lower is considered statistically significant. CI: confidence interval.

Factor		Hazard Ratio	95% CI	*p*-Value
BMA	No	Reference		
	Yes	0.56	0.34–0.91	<0.05
EGFR mutation	No	Reference		
	Yes	0.55	0.34–0.90	<0.05

## Data Availability

The data supporting the findings are available upon reasonable request to the corresponding author.

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
