# Peer review of "Impact of Bone-Modifying Agents on Post-Bone Metastasis Survival Across Cancer Types"

_curroncol, 2025, doi:10.3390/curroncol32010042_

Round 1
Reviewer 1 Report (Previous Reviewer 1)
Comments and Suggestions for Authors
Revision satisfies most of the reviewer's concerns.
Reviewer 2 Report (Previous Reviewer 2)
Comments and Suggestions for Authors
- This second version of the paper is a great improvement, the authors are to be commended.
- The manuscript has been much improved and is in a nice condition now.
Reviewer 3 Report (Previous Reviewer 3)
Comments and Suggestions for Authors
The manuscript entitled: “The effect of bone-modifying agents on post-bone metastasis survival” (ID: curroncol-3389209) by Tamiya et al. aims to investigate whether BMA improves post-bone metastasis survival in different cancer types.
The authors addressed all my initial comments sufficiently.
This manuscript is a resubmission of an earlier submission. The following is a list of the peer review reports and author responses from that submission.
Round 1
Reviewer 1 Report
Comments and Suggestions for Authors
Authors should better define and consider the patient's disease status, (i.e. 1st diagnosis or several relapses, degrees of cancers, presence of any other metastasis than bone).
The time of BMA treatment during cancer progression would also be an important factor that was not described at all.
As the author pointed out limitations, a lack of multivariate analysis significantly raises the question of whether their conclusion will be accepted.
Comments on the Quality of English LanguageOK
Reviewer 2 Report
Comments and Suggestions for Authors
Summary of the Review
Study Design and Manuscript Contents:
・The study is retrospective, involving 539 cancer patients with bone metastasis undergoing rehabilitation, and assesses the impact of bone-modifying agents (BMA) on post-bone metastasis survival.
・The manuscript is structured clearly, with appropriate statistical analyses (Kaplan-Meier, log-rank test, Cox proportional hazards model) and detailed subgroup analyses for different cancers.
Relevant Comments, Major Strengths, and Major Weaknesses:
Strengths:
・The study addresses a critical clinical issue regarding the potential benefit of BMAs for post-bone metastasis survival.
・The large sample size (539 patients) and the clear distinction between different cancer types strengthen the conclusions.
・The use of multivariate analysis to identify independent factors is robust and well-executed.
Weaknesses:
・Selection Bias: The inclusion of only patients undergoing rehabilitation may not represent the broader population, leading to potential bias in the outcomes.
Detailed Comments
1. Title:
・The title is clear but could be slightly refined to emphasize the focus on post-bone metastasis survival and the role of BMA treatment, e.g., "Impact of Bone-Modifying Agents on Post-Bone Metastasis Survival Across Cancer Types."
2. Abstract:
・The abstract succinctly captures the essential points of the study. However, it would be helpful to add more detail about the statistical analyses used (e.g., Kaplan-Meier, multivariate Cox analysis) to provide readers with a clearer understanding of the methodological rigor.
・The statement on the cancer types with poor prognosis (GI and NSCLC without EGFR mutation) should be included to highlight key findings.
3. Introduction:
・Well-written and sets up the context of bone metastasis and its impact on prognosis. It clearly introduces the importance of BMAs in the treatment of skeletal-related events (SRE).
・Consider including more about the challenges or limitations in current BMA treatment approaches, including possible side effects (e.g., MRONJ or AFF), to provide a more balanced context.
4. Materials and Methods:
・The study design and patient inclusion criteria are clearly explained. The authors should clarify how the rehabilitation factor is controlled, especially given its potential impact on survival and the treatment outcomes.
・The use of statistical methods is appropriate. However, more details on the Cox model would be beneficial to enhance transparency, including how potential confounding variables were accounted for.
5. Results:
・The results are well-organized, and the use of Kaplan-Meier curves and multivariate analysis strengthens the findings. However, the table captions could be more descriptive, explaining the significance of certain statistical findings (e.g., p-values, hazard ratios).
・The results of survival for different cancer types and their association with BMA are clear, though more attention could be given to how cancer aggressiveness might influence these outcomes.
6. Discussion:
・The discussion provides a good analysis of the implications of the findings. It emphasizes that BMAs are effective in cancers with poorer prognosis, such as GI and NSCLC (without EGFR mutations).
・More emphasis on the clinical implications of this study in terms of treatment guidelines would be beneficial.
・Consider expanding on the possible biological mechanisms underlying the observed survival benefit with BMA, especially in relation to cancer types like GI and NSCLC.
・Consider discussing the current diagnostic aspects. For instance, you may refer to PMID: 33694240 and PMID: 31192827. While you don’t have to use these articles for discussion, you should find relevant articles to enhance your paper.
7. Conclusion:
・The conclusion is succinct and appropriately reflects the study's findings. It could be strengthened by adding a sentence about the implications for clinical practice, particularly regarding how BMAs should be used in treatment protocols.
8. References:
・The reference list is well-selected, including recent and relevant studies. Ensure that all citations match the journal’s citation style.
・Consider citing more studies that explore BMA's role specifically in cancers with bone metastasis (e.g., more on the thyroid and kidney cancer findings).
9. Figures and Tables:
・The figures (survival curves) and tables (multivariate analysis) are effective in communicating the key results. However, the captions could be more comprehensive, especially for the statistical data.
・The authors should consider adding a figure that demonstrates the proportion of patients receiving different types of treatment alongside BMAs to give a fuller picture of treatment regimens.
Reviewer 3 Report
Comments and Suggestions for Authors
The manuscript entitled: “The effect of bone-modifying agents on post-bone metastasis survival” (ID: curroncol-3389209) by Tamiya et al. aims to investigate whether BMA improves post-bone metastasis survival in different cancer types.
Albeit the manuscript is well written and of special interest, comments should be addressed to further improve the manuscript.
Comments:
1. Table 1: please clarify if p-value is overall measured and not within each cancer type? Moreover, please clarify why 95% CI upper range was in parts N/A and highlight deeper why no HR were shown in the univariate analysis.
2. Figures should be enlarged, to make it more readable.
3. Method section: please clarify the choice of different potential risk factors and why not disease-related and/or therapy-related risk factors were included within uni-and multivariate analysis.
4. Table 2: The presentation of the multivariable analysis should be more clarified for the reader.
- Cancer type: most of the cancer types lower the risk of post-bone metastasis survival, albeit only breast, prostate und thyroid cancers were of significance. Please clarify how “others” within cancer types were defined. The authors should rather enclose uni- and multivariable analysis within one table. In addition, it is unclear why cancer type biliary tract is the reference.
5. Discussion section: the discussion section should be balanced according to strengths and weaknesses of the study. Moreover, please define how future studies could address the limitations of the study. In addition, what can be learnt from this study. What is the exact recommendation for the clinicians.
6. Figure 4 B: since the curves for “no” versus “yes” BMA were crossing, it is not clear why this is highly significant. Please clarify.